# The Evaluation and Key-Factor Identification of the Influence of Tourism on the Soil of Mount Tai

**Fang Li** [1]**, Kailai Wang** [2]**, Xin Li** [1]**, Haodong Zhang** [1] **and Ying Li** [1,*]

1   College of Economics and Management, Shandong Agricultural University, Tai'an 271018, China
2   China National Administration of Coal Geology, Beijing 100038, China
*   Correspondence: li_ying@sdau.edu.cn

**Abstract:** Tourism has been proven to disturb the soil environments and stimulate heavy metal accumulation in scenic areas. Currently, research on the driving mechanisms of the impact of tourism on soil quality degradation is limited. Therefore, the aim of this study was to introduce a complex network method to comprehensively depict the impact of tourism on soil quality. To explore the key influencing factors, we collected 10 topsoil samples and 10 corresponding control samples from representative scenic areas in Mount Tai. Soil physicochemical properties (organic matter (OM), alkali dispelled nitrogen (AN), total nitrogen (TN), available phosphorus (AP), available potassium (AK), electrical conductivity (EC)), heavy metal (Cr, Cd, Pb, Hg, As, Cu) content, and microbial community diversity (by Eco-plate method) were analyzed. Additionally, complex networks of soil quality variables were established based on Pearson correlation coefficients. The results show that the OM, AN, and AP contents of scenic areas are 1.2, 1.03 and 1.18 times higher, while the AK content is 0.97 times lower, than those of the control sites, respectively (mean values of samples from scenic area vs. contorl sites). The single factor index of Hg, As, Pb, Cd, Cu, and Cr are increased from 3.65, 0.53, 0.85, 1.25, 0.78 and 0.58 to 3.69, 0.57, 1.24, 1.75, 0.97, 0.63 and 3.19, respectively, which means that tourism significantly exacerbates soil heavy metal accumulation. Additionally, the soil microbial activity and diversity are slightly reduced due to tourism. In general, the difference in the mean value of each soil quality variable between the scenic spot and the control site is not significant. However, tourism significantly reduced the connectivity and integrity of soil quality variables, which ultimately destabilized the soil, inferred from the comparison of the network's topological parameters. Therefore, raising the content of soil OM and AN and controlling Cd pollution should be given more priority in soil ecosystem protection to counteract the negative impact of tourism on Mount Tai. It was the major limitation of the study that few typical scenic spots were selected as sampling points on Mount Tai. However, this study is sufficient to show that the complex network approach can be extended to other similar studies of soil quality degradation driving mechanisms.

**Keywords:** soil physicochemical property; heavy metals; microbial functional diversity; complex network; tourism impact

## 1. Introduction

The disproportionately high influx of tourists has already affected soil ecosystems, especially in sensitive natural and cultural heritage reserves. Many cases have been reported to be damaged unduly by tourism including ocean and sea shores [1], tropical forests and wetlands [2], and mountain areas. The impacts of tourism on ecosystems are diverse and complex, and the representative threats are supposed to be changes in soil physicochemical properties due to trampling and vegetation degradation [3], and pollutants immission, especially heavy metals (HMs) [4].

As the preferred destination of great tourist influx, the mountain area has gradually increased its position in the international tourism industry [5]. However, mountain soil is very sensitive and fragile, featured with unique topographic conditions, complex vegetation

counterparts, capricious climate, tourism seasonality, and other factors. Furthermore, soil nutrient loss, a chain reaction of soil compaction and erosion, has been demonstrated to be one of the most serious and difficult problems in mountain areas [6]. Likewise, HMs contributed by humans and natural resources accumulate in soils through atmospheric transport, deposition, leaching, and chelation, etc., which are affected by the mountain topography and hydrology. On one hand, tourism aggravates HM pollution (through domestic waste, electronic waste, and the release of sole particles of tourists), on the other hand, it disturbs and changes the distribution of HMs. Memoli et al. [7] indicated that, in addition to HMs deriving by the parent material weathering processes, surface soils of remote or protected areas were also exerted by excessive tourism and received pollutants in gaseous or aerosol forms. Therefore, the assessment of HM pollution is a basis for ensuring the sustainable ecological function of scenic areas. In general, although the impact of tourism on the soil environment has attracted the attention of scholars, the current research has mainly focused on the study of the unilateral effects of tourism on soil nutrient loss or heavy metal pollution. How to comprehensively assess the impact of tourism on soil quality, deeply explore the driving mechanisms of soil quality degradation, and on this basis, seek efficient ways to cope with such degradation is an urgent problem.

Soil microbial community, closely related to soil fertility and decomposition of organic matter, is an important component of ecosystem and soil quality. It also plays an important role in the nutrient cycle, soil structure, degradation of toxic chemicals, and plant pest control [8]. Meanwhile, soil microbes and their participation in biochemical processes are quite sensitive to the dynamics of soil physiochemical properties and soil pollutants, which can be used as a direct indicator of soil quality [9,10]. Soil microorganisms generally exhibit reduced biomass, metabolic activity, number of species, and diversity affected by HMs [11]. Additionally, the microbial community structure will change, due to the adaptive response of part- resistant organisms [12]. Analyzing the function stability and diversity of soil microorganisms can provide a theoretical basis for estimating the availability and health situation of soil resources [13].

Although a large number of previous studies have focused from different angles on soil nutrient succession, HMs spatiotemporal variation, etc. in scenic areas. There are complex correlations between soil physicochemical properties, HM concentrations, and microbial diversities. It is difficult to comprehensively evaluate the negative impacts of tourism on soil quality by studying the trends of only a few variables, let alone the identification of key indicators of soil affected by tourism. As a typical statistical method, the complex network method can be used to characterize information within complex systems, as well as determine the interactions of individual variables and their infuence on the characteristics of the whole network system. The method has flexibility and generality in deeply mining the internal laws of complex natural systems, so it has been widely used in many fields, such as transportation, the Internet, and the power grid [14]. Soil is also a complex system with interrelated quality variables (physicochemical properties, pollutant concentrations, microbial activities, etc.), and it is difficult to determine the degree of influence of external behaviors only by studying changes in independent variables. Additionally, the application of complex network methods in soil science has been explored in recent studies. Zhang et al. [15] determined the key factors that influence soil organic carbon stock in opencast coal-mine dumps based on complex network methods. Li et al. [16] revealed the reconstruction mechanism of reclaimed soil and identified the key indicators of soil reconstruction at different ages by the complex network method. However, this method has not been applied to study the intrinsic changes of soil quality indicators under external influences. Therefore, a complex network method is used in this study for in-depth excavation to comprehensively evaluate the impact of tourism on soil and identify key indicators, which can further provide guidance for soil conservation in tourist areas.

Mountain Tai (Mt. Tai) is the first case in China to be included in the World "Natural and Cultural Heritage" list by UNESCO (United Nations Educational Scientific and Cultural Organization), which is the highest mountain in the North China Plain. Mt. Tai Observatory

is the first permanent alpine meteorological station in China, which has irreplaceable research value in air quality detection. The geographical location and altitude of Mt. Tai make it representative of the regional background air in the North China Plain region [17]. Additionally, the diverse geographical environment and climatic conditions of Mt. Tai provide important habitats for various plants and animal species, so it also has a very important ecological research value. Therefore, it is of multiple types of significance to study the risk of soil damage from tourism on Mt. Tai. However, due to its complex geological structure and topography, the variety of rock types and soil-forming conditions leads to large fluctuations in the content of HMs in soil and high enrichment in local areas [18], in addition, the complex atmospheric and hydrological flow direction lead to the migration, secondary distribution and accumulation of HMs [19,20]. This makes it difficult to study the impact of tourism. Specialists have proved that tourism inevitably brings problems such as soil nutrient loss and heavy metal pollution [21,22]. Currently, only a few references in Chinese have investigated independent soil quality variables and demonstrated that the soil in Mt. Tai was affected by tourism [23,24]. Therefore, it is necessary to use a more scientific and systematic method to identify and evaluate the risks brought by tourism to the soil of Mt. Tai.

In this study, Mt. Tai was selected as our research object. The primary goals are to (1) map the differences in physicochemical properties, HMs contamination, and microbial community diversity in the surface soil of different scenic areas and their corresponding control sites; (2) analyze the correlation between the microbial community and soil environmental factors; and (3) construct complex networks of soil quality variables and preliminarily explore the key soil indicators impact by tourism. This study will guide the selection of efficient manual interventions for soil conservation in Mt. Tai and provide references for the soil's ecosystem management in similar tourist destinations. It was the major limitation of the study that few typical scenic spots were selected as sampling points on Mount Tai. However, this study is sufficient to show that the complex network approach can be extended to other similar studies of soil quality degradation driving mechanisms.

## 2. Materials and Methods

### 2.1. Study Area

Mt. Tai, both natural and cultural heritage, is regarded as one of the most important historical mountains in China and intense human activity can be dated back to 3000 years ago. Mt. Tai is surrounded by three nearby cities, i.e., Tai'an, Ji'nan, and Zibo, which extend 200 km from east to west and 50 km from south to North. The highest point of Mt. Tai is 1545 m above sea level, and the geographic coordinates of the Jade Emperor peak (main peak) are 117°6′ E and 36°16′ N.

This region has a typical continental monsoon climate, which is characterized by hot, humid summers and generally cold, windy, dry winters. Both temperature and precipitation vary significantly with altitude. The vegetation in Mt. Tai is a dominantly warm temperate deciduous broad-leaved forest. The annual precipitation at the mountaintop exceeds 1000 mm, and the annual temperature is between 4 °C to 6 °C, which is significantly different from the foothills (680 mm and 12.8 °C). The soils are mainly brown earth, mountain dark brown earth, and mountain meadow soil, and the soil type changes with altitude which shows a regular zonal distribution from piedmont to summit.

### 2.2. Soil Sampling

Ten representative soil samples (A1–A10) were collected from ten scenic spots along the tourist route in Mt. Tai (Figure 1). Additionally, ten corresponding control samples (B1–B10) were collected at the same altitude and 200 m away from the above A1–A10 samples. Each location was referenced using GPS to determine the longitude and latitude. Surface soil (0–20 cm) was sampled from five to ten sites and then homogenized to make one sample. Subsamples were stored in labeled bags and brought back to the laboratory for analysis. After removing roots and debris, each soil sample was mixed well and

divided into two parts. One part of each sample was air-dried, ground, and passed through a 2 mm nylon sieve for analysis of physicochemical properties and HM concentration. The remaining parts of the samples were stored in a 4 °C refrigerator for soil microbial Biolog-Eco analysis. To reduce the effect of detection error, each sample was measured three times.

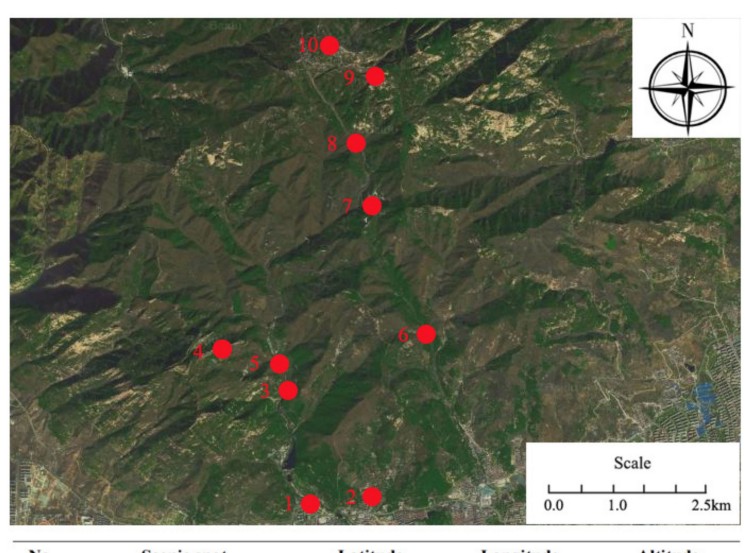

| No. | Scenic spot | Latitude | Longitude | Altitude |
|-----|-------------|----------|-----------|----------|
| 1 | Tian Wai Cun | N36"12"26.4 | E117"06"18.1 | 221m |
| 2 | Pu Zhao Temple | N36"12"30 | E117"06"38 | 210m |
| 3 | Longevity Bridge | N36"13"14.2 | E117"06"105 | 371m |
| 4 | Shan Zi Ya | N36"13"23.8 | E117"05"35.1 | 656m |
| 5 | Wu Ji Temple | N36"13"20.7 | E117"06"08.5 | 383m |
| 6 | Shui Lian Dong | N36"13"33.2 | E117"07"13.7 | 467m |
| 7 | Zhong Tian Gate | N36'14"26.8 | E117"06"54.4 | 813m |
| 8 | Dui Song Shan | N36"15"01.2 | E117"06"31.7 | 1119m |
| 9 | Wang Hai Stone | N36"15"22.5 | E117"06"31.9 | 1485m |
| 10 | Zhang Ren Peak | N36"15"27.2 | E117"06"16.4 | 1424m |

**Figure 1.** Distribution map of the ten scenic spots sampled.

*2.3. Sample Analysis Methods*

Soil pH and electrical conductivity (EC) were measured after oscillation and centrifugation (soil/distilled water ratio (*W/V*) pH:1:2.5; EC:1:5) using a STARTER 300 pH glass electrode (OHAUS Co., Parsippany, Morris, NJ, USA) and a BEC-6600 m electrical conductivity meter (BELL Analytical Instruments Co. Ltd, Dalian, Liaoning, China), respectively. The soil organic matter (OM) content was measured by the potassium dichromate method (refer to the China National Standard GB9834-88). The available phosphorus (AP) was extracted with 0.5 mol·L$^{-1}$ NaHCO$_3$ and the content was measured by Mo-Sb Anti-spectrophotometer. The total nitrogen (TN) content was measured by Kjeldahl determination (refer to the China Agricultural Industry Standard NY/T 1121.24-2012). The alkali-dispelled nitrogen (AN) content was measured by the alkaline hydrolysis diffusion method [25]. The available potassium (AK) content was determined by a FP-640 flame photometer (Shanghai Inesa Scientific Instrument Co. Ltd, Shanghai, China)) after extraction with 1 mol·L$^{-1}$ NH$_4$ OAc ammonium acetate.

The content of HMs (Hg, As, Pb, Cd, Cu, Cr) in soil was measured according to the "Soil Environmental Quality Standards" of China (GB15618-1995), in which the detection limits for Hg, As, Pb, Cd, Cu, and Cr are more than 0.004, 0.100, 0.060, 0.005, 1.000, and 1.000 mg/kg. The test results of all samples are within the range of the detection limits.

Biolog-ECO plate (Biolog Inc., Hayward, CA, USA) was used for microbial community diversity analysis. The detailed steps for extraction, dilution, and inoculation of soil solution are referred from a previous study [8]. The absorbance of each well color was evaluated at 590 nm after 0, 4, 24, 48, 72, 96, 120, 144 and 168 h of incubation, respectively, in darkness at

25 °C. The metabolic activity of each sampling point began to stabilize following incubation at 120 h. Therefore, the average value of absorbance at 120 h, 144 h, and 168 h for all the sampling points were used for calculations and statistical analyses.

### 2.4. Data Analysis

2.4.1. HM Contamination Assessment Methods

Single factor index $P_i$ and Nemerow multi-factor index $P_N$ were calculated according to Equations (1) and (2), respectively, to assess the contamination level of HMs in soils [18].

$$P_i = C_i / S_i \tag{1}$$

$$P_N = \sqrt{\frac{P_{im}^2 + P_{ia}^2}{2}} \tag{2}$$

where $C_i$ represents the measured concentration of the metal $i$ and $S_i$ is the soil environment quality standard (class I) of the corresponding metal $I$; $P_{im}$ and $P_{ia}$ are the maximum and the average of all the metal's relative enrichment $P_i$ at each site, respectively.

2.4.2. The Soil Microbial Diversity Determination Methods

Average well color development (AWCD) indicates the overall carbon source utilization capacity of soil microorganisms, which generally reflects the metabolic activity of the microbial community [26]. The Shannon index ($H$), Simpson index ($D$), and McIntosh index ($U$) are used to assess the functional richness, dominance, and homogeneity of species in the microbial community, respectively [27]. The calculation formula of ACWD and the above three indices are shown in Table 1.

**Table 1.** The calculation formula of ACWD, Shannon index ($H$), Simpson index ($D$), and McIntosh index ($U$).

| | Equation | Specification | Ref. |
|---|---|---|---|
| AWCD | $AWCD = \sum (C_i - R)/31$ | $C_i$ and $R$ are the absorbance values of the $ith$ hole and the control hole | |
| Shannon index | $H' = -\sum P_i \times \ln P_i$ | $P_i$ is the ratio of the relative absorbance of the $ith$ well to the sum of the | [28,29] |
| Simpson index | $D = 1 - \sum P_i^2$ | relative absorbance of the entire plate $(C_i - R)/\sum (C_i - R)$ | |
| McIntosh index | $U = \sqrt{\sum n_i^2}$ | $n_i$ is the relative absorbance of the $ith$ hole $(C_i - R)$ | |

2.4.3. Complex Network Analysis of Soil Quality Variables

Weighted complex networks of scenic spots and control sampling points were built using 18 soil quality variables (7 soil physicochemical variables, 4 microbial diversity indices, 6 heavy metal concentration variables, and altitude data) as nodes, with Pearson correlations as the edge weight connecting any two variables. Four network parameters including node degree, weighted node degree, average path length, and weighted clustering coefficients are used to analyze the soil variable networks. The meanings, interpretations, and calculation methods of the above network parameters refer to our previous literature [16].

The statistical software package SPSS (Statistical Program for the Social Sciences, release 21.0) was used for data processing and correlation analysis. Python Networkx and Origin 2022 were used in complex network parameter calculation and graphical display. All variables follow the normal distributions (tested with the Shapiro-Wilk test at the *p*-value of 0.05, listed in Supplementary Table S1).

## 3. Results and Discussion

### 3.1. Description of Soil Quality Variables

3.1.1. Soil Physicochemical Properties

The soil physicochemical properties are presented in Table 2 (the corresponding standard deviation of each variable is listed in Supplementary Table S2). The soil pH range of scenic spots and control points are 6.26–7.46 and 6.31–7.12, with the standard deviation

values of 0.36 and 0.23, respectively. All sampling points are close to a neutral pH, but the value of scenic spots fluctuates to a greater degree, which is possibly due to tourism disturbance. The average content of OM, AN, AP, and AK of the scenic samples and control samples is 9.68%, 231.88 mg/kg, 54.33 mg/kg, 343.23 mg/kg, and 11.66%, 253.58 mg/kg, 64.27 mg/kg, 336.76 mg/kg, respectively. These soil nutrient variables of both the scenic samples and the control samples are much higher than the background values from the Chinese second national soil survey, which is very likely related to abundant withered trees and fallen leaves [30]. Compared with data from the control sites, the OM, AN, and AP content of scenic areas is 1.2, 1.03, and 1.18 times higher, respectively, while the AK content is 0.97 times lower. It indicates that the surface vegetation in the scenic area is severely destroyed by tourism activities, resulting in the reduction of litter return and OM accumulation [31]. Another reason is that the soil in the scenic area is hardened and compacted by tourists' trampling, which accelerates the OM mineralization process. Reduced OM input and increased mineralization directly lead to lower soil nitrogen content and AP concentration. On the contrary, the scattered incense ash, domestic furnace ash, and other garbage contain a large amount of potassium and phosphorus elements, which may be the main reason for the increase of soil AK content in scenic areas [32].

**Table 2.** Physicochemical properties of soil in different sites.

| Sites | pH | EC | OM (%) | AN (mg/kg) | AP (mg/kg) | AK (mg/kg) | TN (g/kg) |
|---|---|---|---|---|---|---|---|
| A1 | 6.70 | 148.7 | 12.78 | 108.77 | 50.39 | 239.28 | 1.09 |
| A2 | 6.26 | 84.37 | 10.65 | 282.56 | 41.14 | 314.82 | 2.52 |
| A3 | 7.04 | 110.07 | 10.06 | 307.03 | 38.77 | 200.55 | 2.72 |
| A4 | 7.42 | 102.43 | 10.63 | 206.17 | 79.74 | 290.23 | 2.33 |
| A5 | 6.87 | 68.60 | 9.53 | 184.91 | 62.53 | 400.55 | 2.04 |
| A6 | 6.61 | 64.40 | 8.04 | 282.58 | 51.32 | 94.55 | 2.35 |
| A7 | 7.20 | 128.63 | 11.38 | 274.56 | 68.99 | 451.27 | 3.09 |
| A8 | 7.46 | 139.10 | 12.33 | 310.00 | 39.96 | 420.69 | 3.15 |
| A9 | 6.93 | 121.33 | 6.69 | 283.30 | 43.27 | 414.57 | 3.23 |
| A10 | 7.30 | 113.60 | 4.68 | 231.88 | 67.19 | 343.23 | 2.10 |
| Average (A) | 6.98 | 108.12 | 9.68 | 247.18 | 54.33 | 316.97 | 2.46 |
| S.d. (A) | 0.36 | 27.03 | 2.41 | 60.86 | 13.65 | 107.60 | 0.61 |
| B1 | 6.67 | 150.11 | 13.11 | 185.66 | 58.74 | 216.89 | 1.11 |
| B2 | 6.31 | 120.2 | 12.78 | 206.56 | 58.77 | 239.11 | 1.89 |
| B3 | 6.89 | 115.89 | 12.48 | 256.78 | 56.15 | 203.98 | 2.68 |
| B4 | 6.89 | 115.55 | 13.11 | 282.77 | 65.39 | 310.89 | 2.56 |
| B5 | 6.91 | 150.44 | 10.65 | 170.99 | 59.43 | 254.55 | 2.22 |
| B6 | 6.8 | 138.88 | 11.78 | 278.88 | 68.82 | 350.32 | 2.41 |
| B7 | 7.11 | 140.44 | 12.78 | 309.45 | 76.77 | 368.89 | 3.56 |
| B8 | 7.07 | 160.00 | 11.38 | 300.13 | 65.66 | 381.44 | 3.33 |
| B9 | 7.01 | 159.89 | 9.53 | 296.82 | 65.15 | 367.87 | 3.18 |
| B10 | 7.12 | 157.88 | 8.99 | 247.77 | 67.77 | 383.69 | 3.13 |
| Average (B) | 6.88 | 140.93 | 11.66 | 253.58 | 64.27 | 307.76 | 2.61 |
| S.d. (B) | 0.23 | 17.00 | 1.42 | 47.26 | 5.86 | 65.32 | 0.71 |
| B.v. | 6.0-6.9 | - | 9.2 | 57 | 5.3 | 75 | 0.62 |

Abbreviation: Standard devation (S.d.), Background values (B.v.), organic matter (OM), alkali dispelled nitrogen (AN), total nitrogen (TN), available phosphorus (AP), available potassium (AK). Note: The background values of soil fertility referred from the Second National Soil Survey in China, Nutrient data from a sampling point 300m southwest of Zoujialing Village, Jiaoyu Town, Suburban Tai'an City (Soil science database: http://vdb3.soil.csdb.cn/extend/jsp/introduction (accessed on 2 March 2022).

### 3.1.2. Assessment of Soil HM Contamination

The average content of Hg, As, Pb, Cd, and Cu in scenic areas and control sites is all higher than the soil background values of Shandong Province (Supplementary Table S3). Only the average content of Cr is slightly lower than the soil background value (56.88 mg/kg). The environmental risk of the above six HMs was estimated by single-factor index and

Nemerow multi-factor index method. Additionally, the result was listed in Table 3. The accumulation of HMs in the scenic area is significantly higher than that in the control sites (Figure 2). In general, due to the influence of tourism, the HMs in Mt. Tai tend to be enriched. Additionally, it basically shows a trend of increasing with the increase in altitude.

**Table 3.** The single-factor index and the Nemerow multi-factor index.

| Sites | Single Factor Index | | | | | | Nemerow Multi-Factor Index | Pollution Degree |
|---|---|---|---|---|---|---|---|---|
| | Hg | As | Pb | Cd | Cu | Cr | | |
| A1 | 0.26 | 0.64 | 0.84 | 1.10 | 0.52 | 0.59 | 0.91 | Even cleanness |
| A2 | 0.61 | 0.46 | 0.86 | 1.10 | 0.70 | 0.55 | 0.93 | Even cleanness |
| A3 | 0.93 | 0.42 | 0.90 | 1.40 | 0.73 | 0.55 | 1.15 | Slightly polluted |
| A4 | 0.53 | 0.46 | 0.90 | 1.10 | 0.82 | 0.57 | 0.93 | Even cleanness |
| A5 | 1.00 | 0.46 | 0.85 | 1.80 | 0.75 | 0.52 | 1.42 | Slightly polluted |
| A6 | 1.31 | 0.49 | 2.04 | 2.65 | 1.04 | 0.58 | 2.10 | Moderately polluted |
| A7 | 1.07 | 0.49 | 0.99 | 1.90 | 1.21 | 0.68 | 1.54 | Slightly polluted |
| A8 | 10.21 | 0.75 | 1.33 | 2.30 | 1.13 | 0.77 | 7.48 | Strongly polluted |
| A9 | 16.05 | 0.89 | 2.48 | 2.10 | 1.33 | 0.78 | 11.69 | Strongly polluted |
| A10 | 4.93 | 0.70 | 1.25 | 2.00 | 1.42 | 0.70 | 3.72 | Strongly polluted |
| Average (A) | 3.69 | 0.57 | 1.24 | 1.75 | 0.97 | 0.63 | 3.19 | Strongly polluted |
| B1 | 0.27 | 0.64 | 0.83 | 1.10 | 0.52 | 0.59 | 0.91 | Even cleanness |
| B2 | 0.60 | 0.41 | 0.57 | 0.95 | 0.63 | 0.51 | 0.80 | Even cleanness |
| B3 | 0.83 | 0.42 | 0.83 | 1.10 | 0.73 | 0.54 | 0.94 | Even cleanness |
| B4 | 0.53 | 0.41 | 0.72 | 1.30 | 0.75 | 0.55 | 1.05 | Slightly polluted |
| B5 | 0.80 | 0.41 | 0.72 | 1.20 | 0.78 | 0.54 | 1.00 | Even cleanness |
| B6 | 1.20 | 0.49 | 0.79 | 1.15 | 0.85 | 0.55 | 1.04 | Slightly polluted |
| B7 | 1.07 | 0.48 | 0.87 | 1.40 | 0.86 | 0.57 | 1.17 | Slightly polluted |
| B8 | 9.87 | 0.68 | 0.99 | 1.39 | 0.92 | 0.58 | 7.18 | Strongly polluted |
| B9 | 13.19 | 0.68 | 1.11 | 1.46 | 0.86 | 0.69 | 9.56 | Strongly polluted |
| B10 | 8.13 | 0.71 | 1.03 | 1.48 | 0.91 | 0.68 | 5.95 | Strongly polluted |
| Average (B) | 3.65 | 0.53 | 0.85 | 1.25 | 0.78 | 0.58 | 2.96 | Moderately polluted |

Note: The Nemerow multi-factor index $P_N \leq 0.7$, $0.7 < P_N \leq 1.0$, $1.0 < P_N \leq 2.0$, $2.0 < P_N \leq 3.0$, $P_N \geq 3.0$ correspond to clean, even cleanness, slightly polluted, moderately polluted, and strongly polluted, respectively.

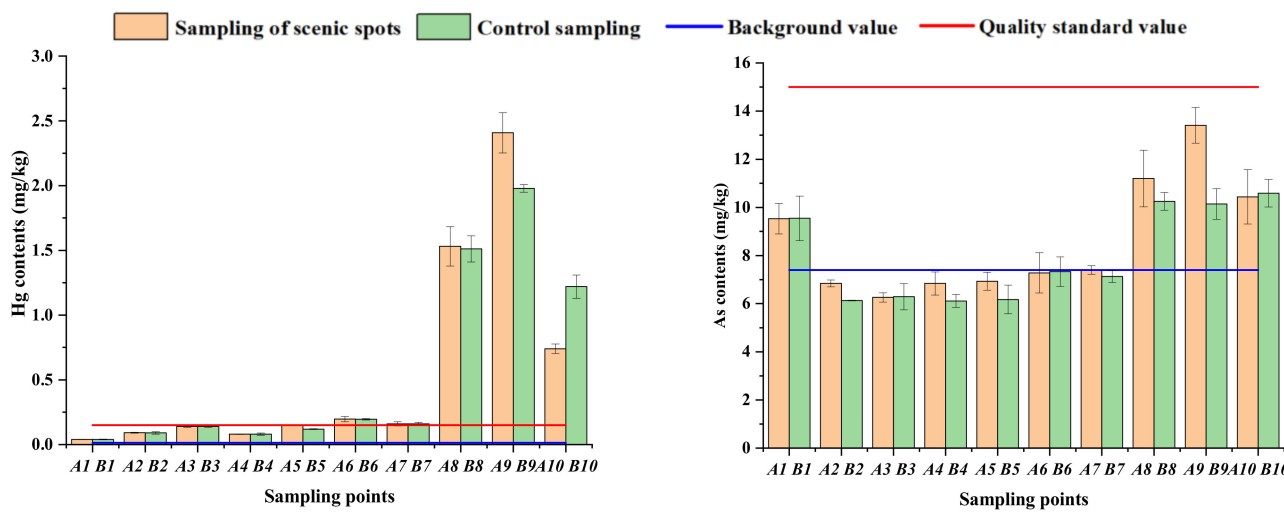

**Figure 2.** *Cont.*

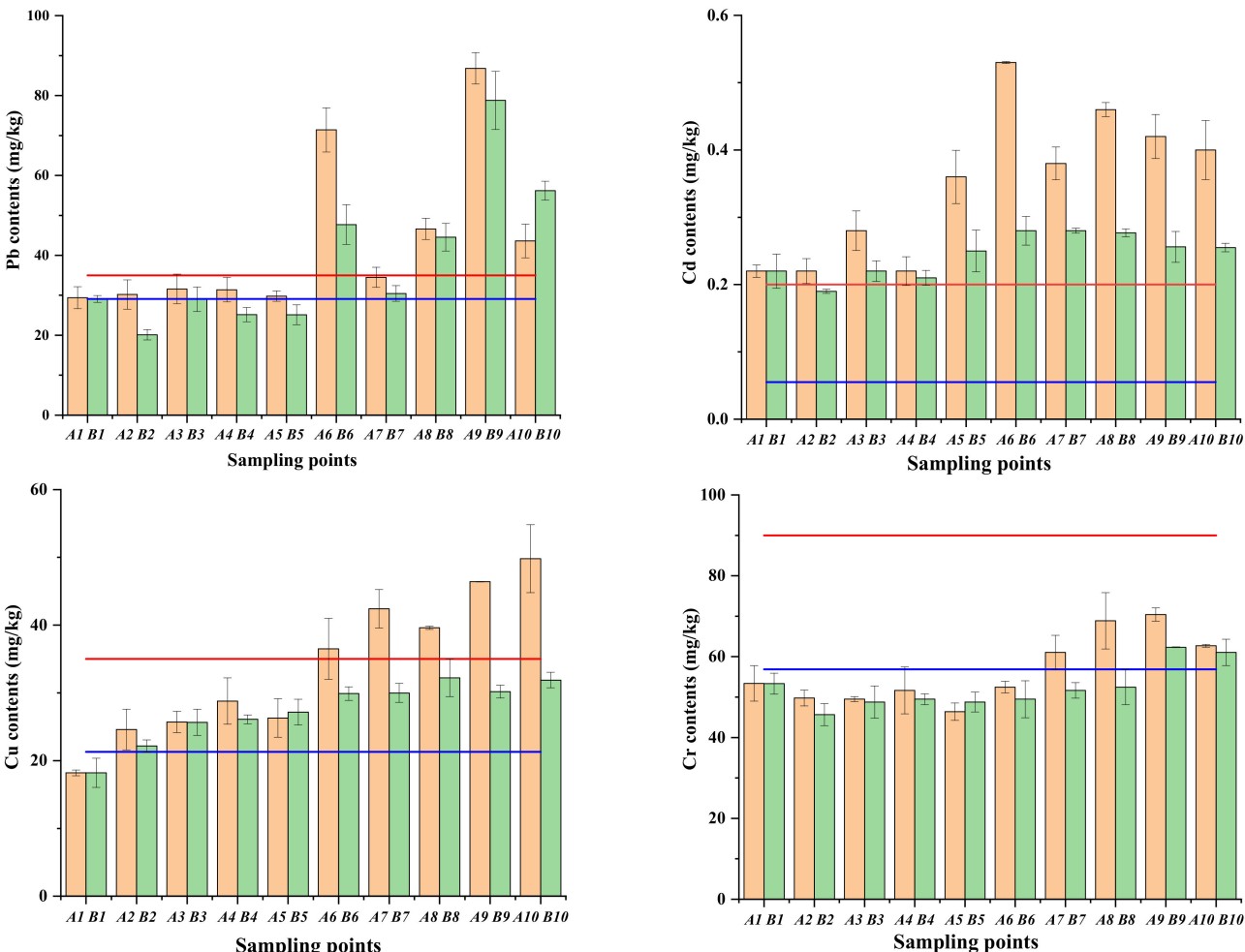

**Figure 2.** Heavy metal concentration in different sampling points. Note: Error bars donate S.D. Quality standard values are referenced from the first-level standard stipulated in "Soil Environmental Quality Standards" of China (GB15618-1995).

By comparing the heavy metal content of samples from the scenic areas and the control sites at the same altitude, it can be seen that in low-altitude areas, there is little difference between them. The result may be attributed to the fact that in low-altitude areas, soil heavy metal pollution caused by tourism is not worth mentioning compared with the heavy metal input from urban industry and transportation. However, in high-altitude areas, the HM concentration of scenic areas is almost all higher than those of control sites, which is most likely caused by tourism activities. For scenic areas, Hg, Pb, and Cd are the three major HMs pollutants in Mt. Tai, since their average single standard indices are greater than one. Among them, the distribution of Hg and Pb is significantly different at the ten sampling sites, which are mainly enriched in high-altitude areas. While the distribution of Cd has no significant difference. The maximum values of Hg and Pb were all observed in the surface soil of sampling point A9 ("Wang Hai Stone", the highest of the ten sampling points with an altitude of 1485 m). Additionally, these maximum values also appear at the control sampling point B9. It is assumed that the cold trapping effect (HMs enrichment by decreasing temperature) dominates the accumulation of Hg and Pb at high altitudes [33–35].

In order to further explore the HM distribution rule in the soil of Mt. Tai and the influence degree of tourism on this pattern, three groups of correlation analysis were conducted. The first one is the HM concentration series of scenic areas with their corresponding control sites, the second one is the HM concentration series of scenic areas with the elevation of the

sampling point, and the third one is the HM concentration series of control sites with the elevation of the sampling point (Table 4). The enrichment of 4 HMs (Hg, As, Cu and Cr) is strongly positively correlated with the altitude in both scenic samples and control samples. It is a good indication that these HMs are homologous and subject to anthropogenic factors and mother rock [33,36]. Furthermore, affected by precipitation, orographic, and related meteorological conditions, these HMs also have similar distribution characteristics. Among these four HMs, tourism contributed the most to the accumulation of Cu, followed by Cr, and the effect increased with altitude. However, for As and Hg, the effect of tourism was significant only at a few points in high- altitude areas. The result may be attributed to the larger number of tourists in high-altitude areas (some tourists take the cable car directly to the top of the mountain without going through other scenic areas), and the longer stays of tourists (many tourists choose to eat and stay at the top of the mountain and wait to watch the sunrise or sunset). In addition, the operation of cable car stations may also be the source of these HMs.

**Table 4.** Correlation analysis of heavy metal contents.

|  | Hg (1–2) | As (1–2) | Pb (1–2) | Cd (1–2) | Cu (1–2) | Cr (1–2) |
|---|---|---|---|---|---|---|
| Pearson correlation | 0.964 ** | 0.913 ** | - | - | 0.886 ** | 0.771 ** |
| Sig. (2-tailed) | 0.000 | 0.000 | 0.670 | 0.134 | 0.001 | 0.009 |
|  | Hg1-Al. | As1-Al. | Pb1-Al. | Cd1-Al. | Cu1-Al. | Cr1-Al. |
| Pearson correlation | 0.812 ** | 0.788 ** | - | - | 0.912 ** | 0.888 ** |
| Sig. (2-tailed) | 0.004 | 0.007 | 0.790 | 0.115 | 0.000 | 0.001 |
|  | Hg2-Al. | As2-Al. | Pb2-Al. | Cd2-Al. | Cu2-Al. | Cr2-Al. |
| Pearson correlation | 0.899 ** | 0.720 * | 0.876 ** | 0.921** | 0.769 ** | 0.859 ** |
| Sig. (2-tailed) | 0.000 | 0.019 | 0.001 | 0.000 | 0.009 | 0.001 |

Abbreviation: Altitude (Al.). Note: **. Correlation is significant at the 0.01 level (2-tailed), * Correlation is significant at the 0.05 level (2-tailed).

The "altitude effect" seems invalid for Pb and Cd in scenic areas. Point 9 (Wang Hai Stone) has extremely high Pb content, which is much higher than those of points 8 and 10 at similar altitudes. It can be interpreted that the flow of tourists is a major source of Pb pollution, which is mainly caused by the shedding and accumulation of particles on the soles of tourists and the random disposal of Pb-bearing batteries [22]. Due to the fact that the "Wanghai Stone," also known as "Riguan Peak," is one of the must-see attractions in Mt. Tai since it is the best viewing spot for sunrises and sunsets. This result is the same as the aggregation of the above four HMs (Hg, As, Cu and Cr). Except that the second highest value of Pb and the highest value of Cd are both observed at point No. 6 ("Shui Lian Dong," with an altitude of 464 m), which is the main reason for the failure of the "altitude effect." It could be deduced that the water vapor-mediated wet deposition inputs and sequestration drove the enhanced accumulation of Pb and Cd at this point. Gerdol and Bragazza [37] also highlighted that the hydrological input by cloud water could account for a major fraction (up to over 50%) of atmospheric wet deposition of Pb and Cd.

### 3.1.3. Functional Diversity of Soil Bacterial Communities

The ecological function of soil microbes can be reflected through the diversity of microbial communities. Considering that AWCD reflects the utilization efficiency of a single carbon source and the metabolic activity of microbial communities, AWCD could be regarded as an important indicator of the functional diversity of microbial communities. Higher AWCD values represent the greater metabolic activity of the soil microbial community [38], and vice versa. The AWCD values of all sampling points increased rapidly during the interval of 0–120 h, and tended to be constant after 120 h, indicating that soil microorganisms had strong utilization efficiencies of carbon source substrates, and almost reached the maximum point until 120 h. Shannon, McIntosh, and Simpson index are also widely used to characterize soil microbial diversity. In general, higher values of Shannon,

McIntosh, and Simpson indices indicate richer species, better community uniformity, and better community diversity, respectively. Based on the AWCD results, the metabolic activity of each sampling point began to stabilize from the 120 h. Therefore, the microbial diversity index of 120 h, 144 h, and 168 h for all the sampling points were calculated and the average value was used (Table 5).

**Table 5.** Functional diversity index of soil microbial community.

|  | AWCD | Shannon | McIntosh | Simpson |
|---|---|---|---|---|
| A1 | 1.27 ± 0.033 | 3.02 ± 0.026 | 7.49 ± 0.236 | 19.42 ± 0.640 |
| A2 | 1.43 ± 0.019 | 3.30 ± 0.010 | 8.39 ± 0.072 | 25.95 ± 0.293 |
| A3 | 1.59 ± 0.016 | 3.36 ± 0.007 | 9.64 ± 0.071 | 28.23 ± 0.240 |
| A4 | 1.49 ± 0.021 | 3.27 ± 0.014 | 8.74 ± 0.165 | 25.70 ± 0.508 |
| A5 | 1.69 ± 0.008 | 3.39 ± 0.002 | 9.42 ± 0.045 | 29.22 ± 0.071 |
| A6 | 1.24 ± 0.068 | 3.20 ± 0.047 | 7.75 ± 0.296 | 22.45 ± 0.655 |
| A7 | 1.39 ± 0.017 | 3.33 ± 0.011 | 9.05 ± 0.236 | 27.14 ± 0.376 |
| A8 | 1.46 ± 0.036 | 3.32 ± 0.009 | 9.27 ± 0.442 | 26.99 ± 0.348 |
| A9 | 1.32 ± 0.112 | 3.17 ± 0.008 | 8.09 ± 0.242 | 22.80 ± 0.382 |
| A10 | 1.50 ± 0.015 | 3.33 ± 0.002 | 8.87 ± 0.050 | 27.13 ± 0.046 |
| Average (A) | 1.438 | 3.269 | 8.671 | 25.470 |
| B1 | 1.38 ± 0.015 | 3.11 ± 0.036 | 7.55 ± 0.36 | 22.45 ± 0.355 |
| B2 | 1.38 ± 0.018 | 3.32 ± 0.019 | 8.56 ± 0.052 | 25.95 ± 0.193 |
| B3 | 1.65 ± 0.006 | 3.41 ± 0.027 | 9.64 ± 0.171 | 30.23 ± 0.290 |
| B4 | 1.43 ± 0.011 | 3.41 ± 0.010 | 9.17 ± 0.095 | 28.70 ± 0.108 |
| B5 | 1.64 ± 0.008 | 3.40 ± 0.002 | 9.52 ± 0.105 | 29.27 ± 0.121 |
| B6 | 1.20 ± 0.045 | 3.33 ± 0.027 | 9.15 ± 0.296 | 26.54 ± 0.253 |
| B7 | 1.25 ± 0.028 | 3.52 ± 0.018 | 10.05 ± 0.096 | 28.33 ± 0.198 |
| B8 | 1.52 ± 0.015 | 3.29 ± 0.013 | 9.97 ± 0.142 | 27.61 ± 0.267 |
| B9 | 1.53 ± 0.015 | 3.22 ± 0.009 | 9.09 ± 0.142 | 25.99 ± 0.282 |
| B10 | 1.49 ± 0.006 | 3.30 ± 0.006 | 9.37 ± 0.099 | 27.66 ± 0.146 |
| Average (B) | 1.440 | 3.329 | 8.983 | 27.190 |

The mean values of these microbial diversity indices for the control samples are all greater than those for the scenic samples, which demonstrates that the microorganisms in the control sites have higher physiological metabolic activity, better diversity, and more homogeneity than in the scenic area. Overall, all sampling points have Shannon values greater than three. The Shannon, Simpson, and McIntosh values differ only slightly between all samples, and also with similar variation trends, which indicates that the influence of tourism has a certain degree of influence on soil microorganisms in Mt. Tai, but the effect is not significant due to the adaptive response of microorganisms.

*3.2. Complex Network Analysis*

3.2.1. Correlation Analysis of Soil Quality Variables

Through the analysis of the above descriptive statistics, it can be seen that the difference between the scenic spot samples and control site samples is not significant. Further research is necessary to mine the influence of tourism on the relationship among soil quality variables through complex network calculations. The first step is to investigate the relationships among soil physicochemical properties, HM contents, and soil microbial community diversity, the correlation analysis was carried out (Supplementary Table S4). Any two of the AWCD, Shannon, McIntosh, and Simpson are significantly correlated at both scenic areas and control sites. It further illustrates that after a long-term adaptive response, the scenic soil microbial community has reached a new stable state. In the scenic samples, there are 21 pairs of correlations among soil quality variables (Figure 3), including six pairs among soil microbial variables, one pair between soil physicochemical variables (TN and AN), and the rest are all among different HMs and the altitude. It can be presumed that the soil physicochemical variables are the most significant factors affected by tourism, which lost all relationships with soil microorganisms and HMs [16]. The significant correlation between

HMs can indicate that they have a certain homology [39]. In comparison, the soil quality variables of the control samples are more closely related (59 pairs of correlations) and form a complex network, indicating that the soil state was more stable [40]. Therefore, it is very essential to further explore the impact of tourism on the soil of Mt. Tai to understand the role of soil quality variables in the network, through the calculation and comparison of topological parameters.

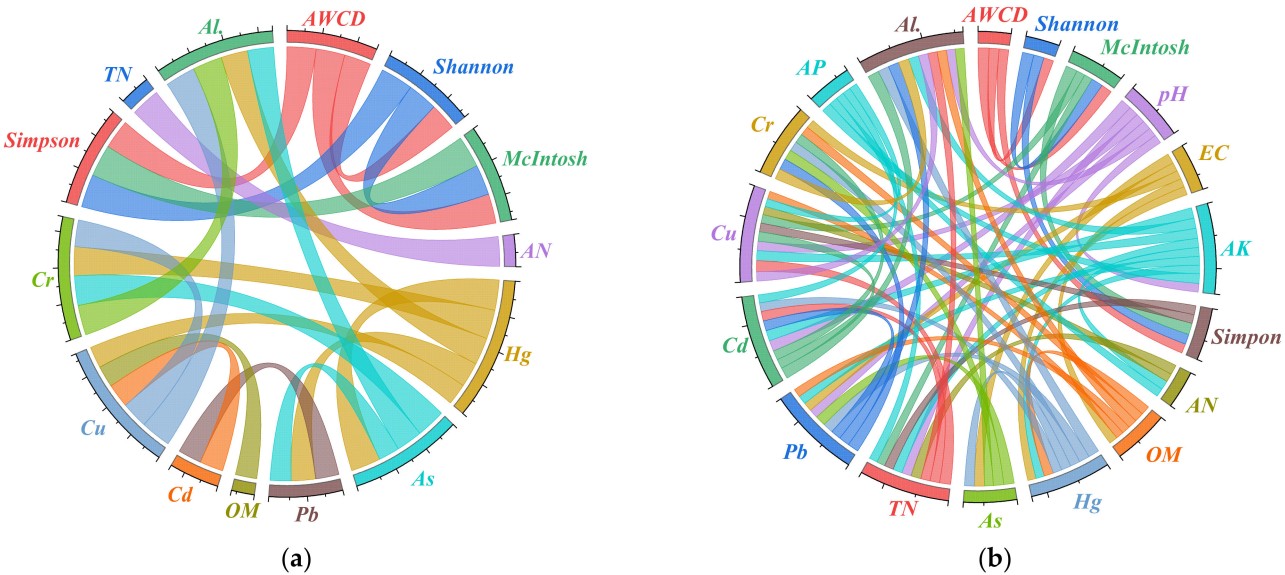

**Figure 3.** Correlation network diagram of soil quality variables for (**a**) scenic area and (**b**) control sites. Abbreviation: Altitude (Al.), organic matter (OM), alkali-dispelled nitrogen (AN), total nitrogen (TN), available phosphorus (AP), available potassium (AK), electrical conductivity (EC).

### 3.2.2. Network Topological Parameters

The network average and weighted average degrees of control sites (6.556 and 5.107,) are more than two times those of scenic areas (3.000 and 2.455). This result means that tourism not only decreases the number of correlations between soil variables but also weakens the connections. The network average path lengths of control sites and scenic areas are 1.457 and 1.941, respectively, indicating that more steps are required for the soil to respond to changes in a soil quality variable. That is, the soil self-regulating ability of the scenic samples has decreased due to the reduced efficiency of network information transmission [15].

The average and weighted average degrees ($k_i$ and $s_i$) and weighted clustering coefficient ($c_i^w$) of each soil quality variable are listed in Table 6. These three parameters respectively represent the number of nodes related to node *i*, the degree of participation of node *i* in the surrounding network, and the connectivity between node *i* and its neighbor nodes [41]. In control sites, altitude and Cu have the largest degree and have first and second weighted degrees, respectively. However, it changed to Hg and Cu in scenic areas. It indicates that the altitude and soil Hg concentration are the key indicators that feature the strongest direct correlation with other soil quality variables in control sites and scenic areas, respectively [42]. Therefore, it can be further inferred that Hg pollution of scenic areas in Mt. Tai presents a dual correlation with mountain deposition and tourism impact. It is important to note, that the results in Table 3 show that the average single-factor index of Hg in the scenic areas is only slightly higher than that in the control sites. We speculate that this might be due to the redistribution of Hg caused by tourism. Zhang et al. [18] also figured that Hg mainly came from artificial sources in Tianshan Mountains, and air-borne depositions are the main source of Hg in soils [43]. This result ties well with previous studies that Hg is unstable in the environment and easily migrates when disturbed [44].

**Table 6.** The topological parameters of soil quality variables.

| | | AWCD | Shannon | McIntosh | Simpson | pH | EC | AK | OM | AN | AP | TN | Hg | As | Pb | Cd | Cu | Cr | Al. |
|---|---|---|---|---|---|---|---|---|---|---|---|---|---|---|---|---|---|---|---|
| $k_i$ | A | 3.00 | 3.00 | 3.00 | 3.00 | - | - | - | 1.00 | 1.00 | - | 1.00 | 5.00 | 4.00 | 3.00 | 2.00 | 5.00 | 4.00 | 4.00 |
| | B | 3.00 | 3.00 | 5.00 | 5.00 | 6.00 | 9.00 | 5.00 | 6.00 | 4.00 | 5.00 | 9.00 | 8.00 | 5.00 | 9.00 | 9.00 | 10.00 | 7.00 | 10.00 |
| $s_i$ | A | 2.56 | 2.70 | 2.73 | 2.81 | - | - | - | 0.65 | 0.88 | - | 0.88 | 3.95 | 3.20 | 2.08 | 1.42 | 3.70 | 3.42 | 3.40 |
| | B | 2.49 | 2.59 | 4.17 | 4.02 | 4.64 | 6.94 | 3.65 | 4.28 | 2.92 | 3.48 | 6.91 | 6.21 | 4.03 | 7.17 | 7.21 | 7.44 | 5.65 | 8.14 |
| $c_i^w$ | A | 1.00 | 1.00 | 0.67 | 0.67 | - | - | - | 0.00 | 0.67 | - | 0.00 | 0.62 | 0.69 | 0.33 | 0.00 | 0.32 | 0.84 | 0.83 |
| | B | 1.00 | 1.00 | 0.50 | 0.52 | 0.87 | 0.90 | 0.65 | 0.68 | 1.00 | 1.00 | 0.49 | 0.69 | 0.90 | 0.62 | 0.60 | 0.42 | 0.77 | 0.59 |

Abbreviation: Altitude (Al.), organic matter (OM), alkali-dispelled nitrogen (AN), total nitrogen (TN), available phosphorus (AP), available potassium (AK), electrical conductivity (EC). Notes: Degree ($k_i$), Weighted degree ($s_i$), Weighted clustering coefficient ($c_i^w$); A represents the network of scenic area samples and B represents the network of control site samples.

The weighted clustering coefficient is a coefficient used to describe the degree of nodes gathering into a small group [45]. Specifically, it is the interconnection level of the adjacent nodes of node *i*, with the range of 0 to 1. Additionally, the nearby nodes tend to be more clustered, when the weighted clustering coefficient of node *i* is closer to 1. The control sites have four extreme pivot nodes (AWCD, Shannon, AN, and AP) with a weighted clustering coefficient value of 1. It reflects that the soil microbial diversity variables and soil nutrient variables have significant small-world network properties. Additionally, there are no nodes with a weighted clustering coefficient of 0. The network of control sites has high information transfer efficiency, that is, the soil self-regulation ability is strong and the soil microbial and nutrient variables and HM concentrations have obvious small-world network layers, and the soil quality state was stable. On the contrary, in the network of scenic areas, the adjacent nodes of OM, TN, and Cd do not form a node pair, showing a weighted clustering coefficient of 0. Therefore, raising the content of soil OM and AN and controlling Cd pollution will be the most direct and effective way to protect the scenic area soil of Mt. Tai [46]. Overall, results demonstrate that tourism obviously damaged soil nutrients, exacerbated HM pollution, and decreased the internal connectivity of soil nutrient variables and HM concentrations, but soil microorganisms remained relatively stable due to their adaptive responses.

## 4. Conclusions

The effects of tourism on soil physicochemical properties, typical HM concentrations, and microbial diversity of scenic areas in Mt. Tai were investigated. In order to explore the influence degree and key factors of tourism on soil, complex networks based on correlation analysis between soil quality variables were established. According to the results, we conclude that: (1) Although tourism reduced the soil OM, AN, and AP content in Mt. Tai from 11.66%, 253.58, and 64.27 mg/kg to 9.68%, 231.88 and 54.33 mg/kg, they were still much higher than the local background value. Conversely, due to the scattering of incense ashes, tourism slightly increased soil AK from 336.76 to 343.23 mg/kg. (2) Affected by deposition, the soil was polluted by HMs to various degrees, especially Hg and Cd, the distribution of which correlated with mountain altitude. Tourism aggravated the enrichment of all HMs, with the greatest impact on Pb, whose average content was increased from 0.85 to 1.24, an approximate 0.46-fold improvement. (3) Tourism slightly reduced soil microbial activity and diversity, however, the effect is not obvious due to the adaptive response of microorganisms. (4) However, it reduced the weighted average degree of the complex networks by more than half and increased the average path length from 1.457 to 1.941, which meant that tourism significantly reduced the connectivity and integrity of soil quality variables and ultimately destabilized the soil. (5) The weighted clustering coefficients of 0 for soil OM, AN, and Cd imply that these three variables are key factors that have the greatest influence on soil integrity and stability. Hence, raising the content of soil OM and AN and controlling Cd pollution will be the most effective way to

protect the soil ecosystem from the impacts of tourism. It was the major limitation of the study that few typical scenic spots were selected as sampling points on Mount Tai. The results provide a theoretical basis and reference for soil protection of Mt. Tai. Given the increasing intensity of tourism and related activities, updated studies are therefore essential to finding more effective ways to minimize the damage to soil ecosystems in the future.

**Supplementary Materials:** The following supporting information can be downloaded at: https://www.mdpi.com/article/10.3390/su142113929/s1, Table S1. Tests of Normality (Shapiro-Wilk); Table S2. Standard deviation of soil physicochemical properties; Table S3. Heavy metal concentrations; Table S4. Correlation analysis of soil quality variables and the altitude.

**Author Contributions:** Conceptualization, F.L.; Formal analysis, K.W.; Funding acquisition, F.L.; Methodology, X.L.; Software, H.Z.; Writing—original draft, F.L.; Writing—review & editing, Y.L. All authors have read and agreed to the published version of the manuscript.

**Funding:** This research was funded by [National Natural Science Foundation of China] grant number 42177027.

**Institutional Review Board Statement:** Not applicable.

**Informed Consent Statement:** Not applicable.

**Data Availability Statement:** Data sharing not applicable.

**Acknowledgments:** The authors would like to acknowledge the editors and reviewers for providing comments and suggestions regarding this paper.

**Conflicts of Interest:** The authors declare no conflict of interest.

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
