# Peer review of "The Evaluation and Key-Factor Identification of the Influence of Tourism on the Soil of Mount Tai"

_sustainability, doi:10.3390/su142113929_

Round 1

Reviewer 1 Report

The paper titled "The evaluation and key-factor identification of the influence of tourism  on the soil of Mount Tai" presents a critical  application of statistical analysis of compositional data. In fact for compositional (constrained) data classical (Euclidean) analysis cannot be applied. Consequently, the application of classical analysis on raw data can seriously affect results and interpretations. Authors would read the wide literature on the argument from 1982 , started with the publication of the book of John Aitchison, updated to now (see also papers and book by Filzmoser P., Pawlowsky-Glahn V., Egozcue J.J., Buccianti A. and so on)

Author Response

Response to reviewers’ comments on “The evaluation and key factors identification of the influence of tourism on the soil of Mount Tai”, (Manuscript ID: sustainability-1972968  ).

The comments of the editor and the reviewers are very much appreciated and helped improve the manuscript significantly. We responded to each of the comments and made all of the suggested changes. Those changes are highlighted with yellow color in the marked revised manuscript which resubmitted as an attachment named “Revised Manuscript with Marks”. In the following section, we explained in details how we responded to each of the comments by repeating the comment and then give a response (blue color) just below it.

Reviewers/Editor comments:

Reviewer: 1

Comments:

The paper titled "The evaluation and key-factor identification of the influence of tourism on the soil of Mount Tai" presents a critical application of statistical analysis of compositional data. In fact for compositional (constrained) data classical (Euclidean) analysis cannot be applied. Consequently, the application of classical analysis on raw data can seriously affect results and interpretations. Authors would read the wide literature on the argument from 1982 , started with the publication of the book of John Aitchison, updated to now (see also papers and book by Filzmoser P., Pawlowsky-Glahn V., Egozcue J.J., Buccianti A. and so on)

Response:

Thank you for your suggestion.

The definition of compositional data is as follows “In statistics, compositional data are quantitative descriptions of the parts of some whole, conveying relative information. Mathematically, compositional data is represented by points on a simplex. Measurements involving probabilities, proportions, and percentages can all be thought of as compositional data.”

Firstly, our original data come from the actual testing of soil-related indicators at the sampling points, which are not compositional data; secondly, in the modeling process of the complex networks, the correlation coefficient between any two indicators is applied as the degree of the line between them, which is also not compositional data. Moreover, such modeling methods have been widely used in previous research (Li et al., 2020; Zhang et al., 2020; Zhang et al., 2022; Zhang et al., 2019).

Reference:

Li, F., Li, X., Hou, L., Shao, A. (2020). A long-term study on the soil reconstruction process of reclaimed land by coal gangue filling. Catena 195, 104874.doi: 10.1016/j.catena.2020.104874.

Zhang, M., Wang, J., Li, S., Feng, D., Cao, E. (2020). Dynamic changes in landscape pattern in a large-scale opencast coal mine area from 1986 to 2015: A complex network approach. Catena 194, 104738.doi: 10.1016/j.catena.2020.104738.

Zhang, X., Li, F., Li, X. (2022). Evolution of soil quality on a subsidence slope in a coal mining area: A complex network approach. Arabian Journal of Geosciences 15(6), 1-13.doi: 10.1007/s12517-022-09815-8.

Zhang, Z., Wang, J., Li, B. (2019). Determining the influence factors of soil organic carbon stock in opencast coal-mine dumps based on complex network theory. Catena 173, 433-444.doi: 10.1016/j.catena.2018.10.030.

Reviewer 2 Report

The paper "The Evaluation and Key-Factor Identification of the Influence of Tourism on the Soil of Mount Tai" falls within the scope of the Sustainability Journal and shows technical relevance.

In this paper, the authors have studied the impact of tourism on soil quality and explore the key influencing factors in representative scenic areas in Mount Tai.

The material is publishable but requires improvement. In this sense, there are some suggestions on the attached paper that should be addressed before publishing.

Suggestion 01

The abstract does not entirely fulfill the function of a summary. This is why the abstract should be reviewed to give, in addition to the method and the main conclusions, the research gap, the aim and the limitations of the work.

Suggestion 02

It is recommended to modify the last keyword by “Tourism impact”.

Suggestion 03

The overall structure of the paper is clear and adequate, however, some key sections, such as the introduction, should be revised. The problem's importance and the study's pertinence must be clearly stated.

Suggestion 04

The limitations of the study should also be included in the introduction section.

Suggestion 05

Novelty unclear: What is the original contribution of the study? The paper is not very enlightening on the subject. Novelty should be made as straightforward as possible.

Suggestion 06

In Figure 3 and Table 6 some abbreviations in the caption are considered to be missing.

Suggestion 07

The conclusion section needs to be also improved. Please paraphrase your results and discussions and use them in the conclusion. The main numerical data must be presented in the conclusion section. The imitations of the study and future lines of research derived from it should also be included in this section.

Author Response

Response to reviewers’ comments on “The evaluation and key factors identification of the influence of tourism on the soil of Mount Tai”, (Manuscript ID: sustainability-1972968  ).

The comments of the editor and the reviewers are very much appreciated and helped improve the manuscript significantly. We responded to each of the comments and made all of the suggested changes. Those changes are highlighted with yellow color in the marked revised manuscript which resubmitted as an attachment named “Revised Manuscript with Marks”. In the following section, we explained in details how we responded to each of the comments by repeating the comment and then give a response (blue color) just below it.

Reviewer 2

The paper "The Evaluation and Key-Factor Identification of the Influence of Tourism on the Soil of Mount Tai" falls within the scope of the Sustainability Journal and shows technical relevance. In this paper, the authors have studied the impact of tourism on soil quality and explore the key influencing factors in representative scenic areas in Mount Tai. The material is publishable but requires improvement. In this sense, there are some suggestions on the attached paper that should be addressed before publishing.

Response:

Thank you for acknowledging this research and for your valuable comments. We responded to all the comments and made all of the suggested changes.

Suggestion 01

The abstract does not entirely fulfill the function of a summary. This is why the abstract should be reviewed to give, in addition to the method and the main conclusions, the research gap, the aim and the limitations of the work.

Response:

The research gap and limitations of the study have been added in the abstract.

Page 1, Line 9-10: “Currently, research on the driving mechanisms of the impact of tourism on soil quality degradation is limited.”

Page 1, Line 10-11:“Therefore, the aim of this study was to introduce the complex network method to comprehensively depict the impact of tourism on soil quality and to explore the key influencing factors.”

Page 1, Line 28-31: “It was the major limitation of the study that few typical scenic spots were selected as sampling points on Mount Tai. However, this study is sufficient to show that the complex network approach can be extended to other similar studies of soil quality degradation driving mechanisms.

Suggestion 02

It is recommended to modify the last keyword by “Tourism impact”.

Response:

Thank you. We have made the suggested changes according to the reviewers' opinions. Page 1, Line 33.

Suggestion 03

The overall structure of the paper is clear and adequate, however, some key sections, such as the introduction, should be revised. The problem's importance and the study's pertinence must be clearly stated.

 Response:

We have added problem's importance and the study's pertinence in the introduction section. Page 2, Line 58-64. “In general, although the impact of tourism on the soil environment has attracted the attention of scholars, the current research has mainly focused on the study of unilateral effects of tourism on soil nutrient loss or heavy metal pollution. How to comprehensively assess the impact of tourism on soil quality, deeply explore the driving mechanisms of soil quality degradation, and on this basis, seek efficient ways to cope with such degradation is an urgent problem.”

Page 2-3, Line 97-99. “Therefore, complex network method is used in this study for in-depth excavation to comprehensively evaluation the impact of tourism on soil and identify key indicators, which can further provide guidance for soil conservation in tourist areas.”

Suggestion 04

The limitations of the study should also be included in the introduction section.

Response:

We have added the limitations of the study in the introduction and conclusion section, Page 3, Line 128-132.

“It was the major limitation of the study that few typical scenic spots were selected as sampling points on Mount Tai. However, this study is sufficient to show that the complex network approach can be extended to other similar studies of soil quality degradation driving mechanisms.”

Suggestion 05

Novelty unclear: What is the original contribution of the study? The paper is not very enlightening on the subject. Novelty should be made as straightforward as possible.

Response:

We have added the description of novelty in the introduction section, Page 2-3, Line 95-99.

“However, this method has not been applied to study the intrinsic changes of soil quality indicators under external influences. Therefore, complex network method is used in this study for in-depth excavation to comprehensively evaluation the impact of tourism on soil and identify key indicators, which can further provide guidance for soil conservation in tourist areas.”

Suggestion 06

In Figure 3 and Table 6 some abbreviations in the caption are considered to be missing.

 Response:

We have added the missing abbreviations in the captions of Figure 3 and Table 6.

“Abbreviation: Altitude (Al.), organic matter (OM), alkali dispelled nitrogen (AN), total nitrogen (TN), available phosphorus (AP), available potassium (AK), electrical conductivity (EC).”

Suggestion 07

 The conclusion section needs to be also improved. Please paraphrase your results and discussions and use them in the conclusion. The main numerical data must be presented in the conclusion section. The limitations of the study and future lines of research derived from it should also be included in this section.

 Response:

We have paraphrased the main results and discussions in the conclusion section, which includes the main numerical data.

Page 14, Line 427: “from 11.66%, 253.58 and 64.27 mg/kg to 9.68%, 231.88 and 54.33 mg/kg,”

Page 14, Line 429: “from 336.76 to 343.23 mg/kg”

Page 14, Line 432-433: “whose average content was increased from 0.85 to 1.24, an approximate 0.46-fold improvement”

Page 14, Line 435-437: “it reduced the weighted average degree of the complex networks by more than half and increased the average path length from 1.457 to 1.941,”

Page 14, Line 438-440: “The weighted clustering coefficients of 0 for soil OM, AN, and Cd imply that these 3 variables are key factors that have the greatest influence on soil integrity and stability.”

The limitations of the study of research have been added in the conclusion section (Page 14, Line 442-443: “It was the major limitation of the study that few typical scenic spots were selected as sampling points on Mount Tai.”).

And future lines of research derived from this study have been included in the manuscript. Page 14, Line 444-446: “Given the increasing intensity of tourism and related activities, updated studies therefore are essential to find more effective ways to minimize the damage to soil ecosystems in the future.”  

Round 2

Reviewer 1 Report

Concentration of chemical elements are compositional data. Correlations between compositional data calculated by using classical correlation coefficients are wrong. The problems of compositional data could be analyzed with more attention.

Reviewer 2 Report

No remaining suggestions are needed